# Competitive Binding of Magnesium to Calcium Binding Sites Reciprocally Regulates Transamidase and GTP Hydrolysis Activity of Transglutaminase 2

**DOI:** 10.3390/ijms21030791

**Published:** 2020-01-25

**Authors:** Eui Man Jeong, Ki Baek Lee, Gi Eob Kim, Chang Min Kim, Jin-Haeng Lee, Hyo-Jun Kim, Ji-Woong Shin, Mee-ae Kwon, Hyun Ho Park, In-Gyu Kim

**Affiliations:** 1Department of Biochemistry and Molecular Biology, Seoul National University College of Medicine, Seoul 03080, Korea; euimanjeong@gmail.com (E.M.J.); lkb3431@gmail.com (K.B.L.); wlsgod@snu.ac.kr (J.-H.L.); thanatos460@snu.ac.kr (H.-J.K.); earthbear3@gmail.com (J.-W.S.); kma1895@gmail.com (M.-a.K.); 2College of Pharmacy, Chung-Ang University, Seoul 06974, Korea; gieob1994@naver.com (G.E.K.); 6427372@naver.com (C.M.K.)

**Keywords:** transglutaminase 2, calcium, magnesium, X-ray crystallography

## Abstract

Transglutaminase 2 (TG2) is a Ca^2+^-dependent enzyme, which regulates various cellular processes by catalyzing protein crosslinking or polyamination. Intracellular TG2 is activated and inhibited by Ca^2+^ and GTP binding, respectively. Although aberrant TG2 activation has been implicated in the pathogenesis of diverse diseases, including cancer and degenerative and fibrotic diseases, the structural basis for the regulation of TG2 by Ca^2+^ and GTP binding is not fully understood. Here, we produced and analyzed a Ca^2+^-containing TG2 crystal, and identified two glutamate residues, E437 and E539, as Ca^2+^-binding sites. The enzymatic analysis of the mutants revealed that Ca^2+^ binding to these sites is required for the transamidase activity of TG2. Interestingly, we found that magnesium (Mg^2+^) competitively binds to the E437 and E539 residues. The Mg^2+^ binding to these allosteric sites enhances the GTP binding/hydrolysis activity but inhibits transamidase activity. Furthermore, HEK293 cells transfected with mutant TG2 exhibited higher transamidase activity than cells with wild-type TG2. Cells with wild-type TG2 showed an increase in transamidase activity under Mg^2+^-depleted conditions, whereas cells with mutant TG2 were unaffected. These results indicate that E437 and E539 are Ca^2+^-binding sites contributing to the reciprocal regulation of transamidase and GTP binding/hydrolysis activities of TG2 through competitive Mg^2+^ binding.

## 1. Introduction

Transglutaminase 2 (TG2) is a calcium-dependent enzyme that regulates many biological processes, including apoptosis, autophagy, inflammation, and extracellular matrix formation [1]. These are mediated by the transamidation activity of TG2, i.e., polyamine incorporation or crosslinking substrate proteins such as caspase-3, IκB, and collagen, thus modulating their activities [2,3]. Intracellular TG2 is inactive [4]. In response to oxidative stress such as UV-irradiation, or treatment with chemotherapeutic agents, TG2 is activated by increasing concentrations of calcium ions (Ca^2+^) [5,6]. Previous reports have shown that aberrant TG2 activation is involved in the pathogenesis of a variety of diseases, such as cancer progression, cataract, and celiac disease. In particular, TG2 activity is critical in the development of fibrotic diseases, including cystic fibrosis, renal fibrosis, and pulmonary fibrosis [3,7].

TG2 is regulated by the binding of guanine nucleotides and Ca^2+^. The transamidation activity of TG2 is inhibited by GTP or GDP binding, which stabilizes its closed structure, thereby obstructing the accessibility of substrates to the active sites [8,9]. In contrast, Ca^2+^ acts as a cofactor in the transamidation activity by inducing conformational change to the active open form [10]. TG2 is capable of binding up to six Ca^2+^ [11,12]. Five putative calcium-binding sites on the catalytic core domain have been predicted based on sequence comparison with the Ca^2+^-bound crystal structure of FXIIIa and TG3 [13,14,15] and an analysis of deeply negative charged sites [12]. However, the exact location of these calcium-binding sites is still unknown.

Magnesium (Mg^2+^) is the most abundant divalent cation, existing in concentrations of 17–20 mM. As most Mg^2+^ is present in complex with ATP, GTP, nucleic acids, and other molecules in cells, free (Mg^2+^) is present at 0.5–1 mM in mammalian cells [16]. Total extracellular (Mg^2+^) is maintained at 0.7–1 mM in humans [17]. Thus, cytosolic free (Mg^2+^) oscillates in a relatively narrow dynamic range, compared to (Ca^2+^), which is present at ~100 nM in the cytosol and 2 mM in the extracellular space [18]. However, Mg^2+^ plays an essential role in regulating broad cellular processes [19,20]. In particular, Mg^2+^ inhibits the Ca^2+^-activated transamidation activity of TG2 [21], but promotes GTP-binding and GTP hydrolyzing activities [22]. In G proteins, Mg^2+^ assists the GTP hydrolyzing activities [23], and usually exists in crystal structures of G proteins in complexing with GTP [24]. Interestingly, Mg^2+^ has not been found in the crystal structures of TG2 in complex with GTP or GDP [8,9]. Thus, the mechanism by which Mg^2+^ inhibits Ca^2+^-activated transamidation activity has not been elucidated.

In the present study, we identified two glutamate residues as Ca^2+^-binding sites by resolving a Ca^2+^-containing crystal of TG2 and investigating the role of these sites in the regulation of TG2, including its transamidase, GTP binding, and hydrolysis activities. Interestingly, our results show that Mg^2+^ competitively binds these Ca^2+^-binding sites, contributing to the reciprocal regulation of transamidase and GTP binding/hydrolysis activities of TG2. Given the relatively high concentration of Mg^2+^ in cells, binding of Mg^2+^ to these Ca^2+^-binding sites is critical for preventing the aberrant activation of intracellular TG2.

## 2. Results

### 2.1. Identification of New Ca^2+^-Binding Sites in TG2

To identify the Ca^2+^ binding sites, a Ca^2+^-containing crystal of TG2 was produced in totally different crystallization conditions from those used for the previous structural study of TG2 [25]. A relatively low-resolution structure of 3.56 Å was solved and refined to *R*_work_ = 22.5% and *R*_free_ = 26.3%. The crystallographic and refinement statistics are provided in Table 1. The structure revealed that the asymmetric unit (ASU) cell contained three molecules. The final structural models were constructed with residues 2-687 for chain A, residues 3-687 for chain B, and residues 9-687 for chain C. GDP was detected at the nucleotide binding site in each chain. After careful analysis by comparing the current structure with previous TG2 structures that do not contain Ca^2+^, we identified two tentative Ca^2+^ binding sites in the structure of TG2. These two tentative Ca^2+^ binding sites are located in the catalytic core domain and in the β-barrel 1 domain (Figure 1A). The first binding site is located in the catalytic domain and is coordinated by the side chains of E437, Y159, and E155. The second binding site is located in the β-barrel 1 domain and is coordinated by the main chains of N497 and F537, and side chains of E539 and N498. The electron density comparison showed that Ca^2+^ electron densities are only detected in the current structure, indicating that the density is produced by Ca^2+^ (Figure 1B,C). Superposition analysis for structural comparison between the TG2/Ca^2+^ complex and the apo-form of TG2 (PDB ID: 1KV3) showed that the two loops formed by amino acids 406–412 and 460–470 were structurally different, indicating that the structure of these loops might be affected by Ca^2+^ binding (Figure 1D).

Sequence comparison of site 1 in the TG family shows that the Tyr residue corresponding to the Ca^2+^-coordinating Y159 of TG2 is conserved in all human TG members, whereas that corresponding to E437 or E155 of TG2 is not. In Site 2, the Asn residue corresponding to N497 is relatively well conserved, whereas that corresponding to N498, F537, or E539 of TG2 is not (Figure 2A). On the contrary, inter-species sequence comparison shows that E155, Y159, and E437 in Site 1 are well conserved in mammals, birds, and frogs, but not in fishes or flies. In Site 2, N497 and E539 are well conserved in vertebrates except red seabream, but not N498 and F537 (Figure 2B), suggesting that these Ca^2+^-binding sites are unique to human TG2.

### 2.2. Ca^2+^ Binding to E437 and E539 Residues Is Required for the Transamidase Activity of TG2

To confirm the Ca^2+^ binding at these residues, we assessed Ca^2+^-activated transamidase activity of TG2 using mutants of E437 and E539 substituted with Arg by site-direct mutagenesis. We first performed circular dichroism (CD) spectroscopic analysis with purified recombinant proteins, wild type (WT), E437R, E539R, and E437/539R to evaluate the effects of mutation on the structural conformation of TG2. CD spectra of E437R, E539R, and E437/539R showed no significant difference compared with the WT, although subtle differences were observed, indicating that the folding properties of TG2 were not affected by mutation (Figure 3A). We then tested whether the affinity of TG2 for Ca^2+^ is affected by Arg substitution of E437 and/or E539 using isothermal titration calorimetry (ITC). ITC analysis showed that the WT has a *K*_d_ of 0.027 μM, whereas E437R and E539R exhibited an approximately 6963- and 1.3-fold higher *K*_d_ for Ca^2+^, respectively. E437/539R did not show an appropriate net heat curve, indicating that Ca^2+^ did not bind to this mutant TG2 (Figure 3B and Table 2). Moreover, transamidase activity assayed with various Ca^2+^ concentrations showed that the half-maximal effective concentration (EC_50_) of Ca^2+^ for E437R or E539R are similar to that of WT, whereas the EC_50_ of E437/539R is about 1.3-fold higher than that of the WT (Figure 3C,D). These data indicate that E437 and E539 are critical residues for Ca^2+^ binding and are required for the full transamidase activity of TG2.

### 2.3. Mg^2+^ Binding to E437 and E539 Inhibits the Transamidase Activity of TG2

Previous reports have shown that TG3 is inhibited by binding of Mg^2+^, instead of Ca^2+^, which closes the channel for the active site [26]. Moreover, the transamidase activity of TG2 is also suppressed in the presence of Mg^2+^ [21]. Thus, to test whether Mg^2+^ could inhibit TG2 by binding to E437 or E539, we measured the transamidase activity of E437R, E539R, and E437/539R with increasing concentrations of Mg^2+^ in the presence of 2 mM CaCl_2_. The enzyme activities of both WT and mutants were inhibited by Mg^2+^, whereas mutants of Ca^2+^-binding sites were less sensitive to Mg^2+^ (Figure 4A), as shown by the significantly higher calculated half-maximal inhibitory concentrations (IC_50_) of Mg^2+^ for E437R, E539R, and E437/539R than that of the WT (9.3, 9.7, and 10.4 mM, respectively, compared with 8.3 mM for the WT; Figure 4B). To confirm the binding of Mg^2+^ to Ca^2+^-binding sites, we compared the binding affinity of Mg^2+^ for WT and mutants. ITC analysis showed that the *K*_d_ of WT TG2 is 0.215 μM, and that the *K*_d_ of E437R and E539R are about 6.8- and 1.3-fold higher than that of the WT, respectively. Moreover, E437/539R did not exhibit a proper net heat curve (Figure 4C and Table 2). These data indicate that Mg^2+^ inhibits transamidase activity by competitive binding with Ca^2+^ to the E437 and E539 residues of TG2.

### 2.4. Mg^2+^-Binding to E437 and E539 Promotes the GTP Binding and Hydrolysis Activity of TG2

Mg^2+^ is required for the inhibition of transamidase by GTP binding as well as GTP hydrolysis activity [22]. To test whether Mg^2+^-binding to E437 and E539 residues could affect GTP binding to TG2, the GTP binding activity was compared between WT and mutants using BODIPY FL GTP-γ-S, which recovers the quenched fluorescence by binding G-proteins. R580A was used as a GTP-binding defective mutant [27]. When incubated with increasing concentrations of MgCl_2_ in the presence of 1.6 μM BODIPY FL GTP-γ-S, the WT showed an apparent sigmoidal curve of GTP binding, but not E437R, E539R, and E437/539R. The maximum GTP binding of E437R, E539R, and E437/539R was approximately 6.8-, 2.7-, and 2.6-fold lower than that of the WT, respectively (Figure 5A). R580A did not exhibit any fluorescence at all concentrations of MgCl_2_ as described in a previous report [27]. Quantitatively, in the presence of 1 mM MgCl_2_, the GTP binding of E437R, E539R, and E437/539R was about 6.7-, 3.7-, and 4.6-fold lower than that of the WT, respectively (Figure 5B). Moreover, in the presence of 1 mM GTP, the GTP hydrolysis activity of E437R, E539R, and E437/539R, measured by the amount of phosphate released from GTP, was about 1.6-, 1.6-, and 1.4-fold lower than that of the WT, respectively (Figure 5C). Under the same experimental conditions, when comparing transamidase activity, WT showed a gradual decrease in transamidase activity with increasing concentrations of GTP, but E437R, E539R, and E437/539R required a higher GTP concentration for similar TG2 inhibition (Figure 5D), as demonstrated by the about 1.4-fold higher IC_50_ of E437/539R than the WT, E437R, or E539R (Figure 5E). These results indicate that binding of Mg^2+^ at E437 and E539 residues allosterically promotes GTP binding and hydrolysis activity, resulting in inhibition of the transamidase activity of TG2.

### 2.5. Mg^2+^-Binding to E437 and E539 Is Critical for Preventing the Activation of Intracellular TG2

The cytosolic concentration of Mg^2+^ is about 10^4^-fold higher than that of Ca^2+^ [16,18], and intracellular TG2 is inactive under normal culture conditions [4]. To test whether Mg^2+^-binding to E437 and E539 residues renders TG2 inactive in cells, we transfected the expression constructs for WT and mutant TG2 (E437R, E539R, E437/539R, and R580A) into HEK293 cells, which express low levels of endogenous TG2, and compared their intracellular transamidase activity. An in situ biotinylated pentylamine (BP) incorporation assay revealed that cells expressing TG2 mutants exhibit significantly higher transamidase activity compared with that of cells expressing WT-TG2 (Figure 6A). We then tested whether the concentration of Mg^2+^ in culture media could affect intracellular TG2 activity. Human dermal fibroblasts (HDF) and HCT116 cells that have high expression levels of endogenous TG2 were cultured in media containing 0, 0.2, 0.4, or 0.8 mM of Mg^2+^ for 2 days (0, 25, 50, and, 100%, respectively, compared to complete media). Transamidase activities in both cells were significantly increased with decreasing Mg^2+^ levels in the culture media (Figure 6B). To confirm these results, we compared the intracellular transamidase activity of cells expressing WT and mutant TG2 cultured in normal and Mg^2+^-depleted media. HEK293 cells expressing WT-TG2 showed about a 2.2-fold increase in enzyme activity, whereas cells expressing the E437/539R mutant showed a 1.2-fold increase in Mg^2+^-free media (Figure 6C). These results explain how TG2 remains inactive in intracellular environments.

TG2 activates the NF-κB signaling pathway that leads to various inflammatory diseases, such as bleomycin-induced lung inflammation and UV-induced skin inflammation, through production of pro-inflammatory cytokines [7,28]. To evaluate the role of Mg^2+^-binding in the regulation of NF-κB signaling, we compared the NF-κB luciferase reporter activity in HEK293 cells transfected with WT and mutant TG2. Cells expressing TG2 mutants, especially E437/539R, exhibited significantly higher levels of reporter activity than cells expressing WT-TG2 (Figure 6D). Moreover, Mg^2+^ depletion in culture media resulted in about a 2.6-fold increase in the luciferase reporter activity of cells transfected with WT-TG2. In contrast, this effect of Mg^2+^ depletion was abrogated in cells expressing E437/539R (Figure 6E). To confirm these results, we compared the effect of Mg^2+^ depletion on the level of p65 phosphorylation in cells expressing the WT and mutant TG2. When cultured in normal media, the levels of p65 phosphorylation in cells expressing mutant TG2 were higher than those in cells expressing WT-TG2. In contrast, when cultured in Mg^2+^-free media, p65 phosphorylation was increased in cells expressing WT-TG2, but not in cells expressing E437R, E539R, or E437/539R (Figure 6F). These results indicate that Mg^2+^-binding to E437 and E539 of TG2 is crucial in preventing aberrant activation of the NF-κB signaling pathway.

## 3. Discussion

We resolved a structure of Ca^2+^-bound TG2 by X-ray crystallography and identified two Ca^2+^ binding sites composed of E437 and E539 as intensely Ca^2+^-coordinated amino acids. Substitution of E437 and E539 with Arg attenuated the Ca^2+^-binding affinity and Ca^2+^-dependent transamidase activity of TG2. Moreover, ITC analysis showed that Mg^2+^ also binds E437 and E539, which is crucial for GTP binding and GTPase activity of TG2. We further showed that E437 and E539 mutants diminish the GTP-mediated inhibition of TG2 transamidase activity, and that Mg^2+^ depletion in culture media highly increases the intracellular transamidase activity of WT-TG2, but not of E437R and/or E539R mutants. Therefore, TG2 is allosterically regulated by the competitive binding of Ca^2+^ and Mg^2+^ to E437 and E539 (Figure 6G).

Kiraly et al. have suggested five Ca^2+^ binding sites (S1–S5) on the catalytic core domain of TG2 by comparative analyses of FXIIIa and TG3 sequences (S1–S3) and by searching highly negative charged surface patches (S4 and S5) [12]. E437 is one of four negatively charged amino acids located in S5. In contrast, E539 is located in the β-barrel 1 domain and has not been suggested as a Ca^2+^ binding site. Given that human TG2 binds six Ca^2+^ [11,12], E539 is a newly identified site for Ca^2+^ binding. Binding of Ca^2+^ to E437 and E539 induced some structural changes in two loops of TG2, amino acids 406–412 and 460–470 (Figure 1D). Interestingly, both loops are located in the hinge region of TG2. Thus, binding of Ca^2+^ to E437 and E539 is likely to promote an open active conformation of TG2 by altering the hinge structure (Figure 6G).

TG2 is reported to bind up to six Ca^2+^ [11,12] that act as cofactors in the transamidase activity by dissociating GTP or GDP and promoting a conformational change to the open form [10]. Intriguingly, the present Ca^2+^ bound TG2 crystal structure is a GDP-bound compact form that binds Ca^2+^ at only two sites, namely, E437 and E539. The E437/539R mutant required higher (Ca^2+^) for the transamidase activity of TG2 than the WT (Figure 3C,D), and showed no Ca^2+^ binding affinity (Figure 3B and Table 2). Therefore, E437 and E539 seem to be the initial Ca^2+^ binding sites in the process of TG2 activation, and Ca^2+^ binding on E437 and E539 appears to be a prerequisite for Ca^2+^ binding at the other sites.

In biological systems, Mg^2+^ can counteract Ca^2+^ by competitive binding to the same site of some enzymes in which despite the same charge and similar chemical reactivity, the conformation and functions are differently regulated by binding of Ca^2+^ or Mg^2+^. For example, the Mg^2+^-dependent human phosphoserine phosphatase (HPSP) is inhibited by Ca^2+^ binding [29]. Crystal structures of HPSP show that Mg^2+^ is coordinated six-fold in the active sites, whereas Ca^2+^ is coordinated seven-fold in the same sites and disrupts a nucleophilic attack on the substrate phosphoserine by capturing all oxygen atoms of catalytic Asp residue. On the contrary, in TG3, simultaneous binding of Ca^2+^ on the S1, S2, and S3 sites opens a channel of the active site for full enzyme activity, whereas binding of Mg^2+^ on the S3 site leads to closing of the channel and limits the accessibility of substrates to the active site [26]. Interestingly, the site of TG2 corresponding to the S3 site of TG3 also seems critical for crosslinking activity through Ca^2+^ binding [12]. Our data showed that mutation of E437 and E539 to Arg was not sufficient to abolish the Mg^2+^-dependent GTP binding to TG2 (Figure 5), suggesting that Mg^2+^ also competitively binds to the S3 site of TG2, facilitating the inactive GTP-bound conformation.

The basal GTP level in cells is about 500 µM [30], and GTP binds to a cleft between the catalytic core domain and β-barrel 1 domain of TG2 [9]. However, there are no serine/threonine residues for binding the phosphates of GTP and coordinating Mg^2+^ [9] unlike other GTP binding proteins such as small G proteins [24,31,32]. Our data show that Mg^2+^ is coordinated to E437 and E539 in TG2, which are apart from the GTP binding site, indicating that TG2 may be allosterically regulated by Mg^2+^ in the process of GTP binding and hydrolysis.

In humans, chronic Mg^2+^ deficiency is implicated in various pathogeneses, including colorectal cancer [33,34,35], pancreatic cancer [36], coronary atherocalcification, and vessel stiffness [37]. In cultured cells, Mg^2+^ depletion promotes the production of cytokines through NF-κB activation [20] and cellular senescence [19]. Intriguingly, it has been observed that abnormal TG2 activation is also associated with upregulation of NF-κB signaling and pathogenesis of various age-related diseases, including cancers and cardiovascular diseases [3]. Our results demonstrated that Mg^2+^ critically regulates the sensitivity of TG2 to Ca^2+^, preventing its aberrant activation. Therefore, it is likely that Mg^2+^ deficiency is a plausible mechanism for aberrant TG2 activation, and that magnesium supplementation could be a preventive and therapeutic regimen for TG2-associated diseases.

## 4. Materials and Methods

### 4.1. Protein Expression and Purification

The method for TG2 protein expression and purification was introduced in previous structural studies [25]. Briefly, the full-length human TG2 gene encoding G224 was inserted into a home-made pOKD5 vector. The plasmid was transformed into BL21 (DE3) *E. coli* competent cells and was expressed by treatment with 0.125 mM isopropyl β-D-thiogalactopyranoside (IPTG) for 25 h at 18 °C. Harvested cells were lysed by sonication in 30 mL lysis buffer (50 mM sodium-phosphate buffer at pH 7.5, 400 mM NaCl, 5 mM benzamidine, 1 mM 2-mercaptoethanol, 50 μM GTP, 1 mM PMSF, 0.5% (v/v) Triton X-100, and 10 mM imidazole). After removing the lysate by centrifugation at 4°C, the supernatant was applied to a gravity-flow column (Bio-Rad) packed with Ni-NTA affinity resin (Qiagen). The unbound impurities were removed from the column using 100 mL washing buffer (50 mM sodium-phosphate buffer at pH 7.5, 400 mM NaCl, 5 mM benzamidine, 1 mM 2-mercaptoethanol, 50 μM GTP, 1 mM PMSF, 0.5% (v/v) Triton X-100, and 20 mM imidazole). The target protein was eluted from the column using elution buffer (50 mM HEPES buffer at pH 7.0, 100 mM NaCl, 50 μM GTP, 10% (v/v) glycerol, and 250 mM imidazole). The eluted TG2 was then applied to a Superdex 200 gel filtration column (GE healthcare) that had been pre-equilibrated with a buffer containing 20 mM Tris at pH 8.0, 150 mM NaCl, and 5 mM CaCl_2_. Purified TG2 was then applied to a mono Q ion-exchange column (GE healthcare) using starting buffer (20 mM Tris pH 8.0) and elution buffer (20 mM Tris, pH 8.0 and 1 M NaCl). The eluted TG2 was finally applied to a Superdex 200 gel filtration column (GE healthcare) that had been pre-equilibrated with a solution of 20 mM Tris at pH 8.0, 150 mM NaCl, and 5 mM CaCl_2_. The purified TG2 was collected and concentrated to 10–11 mg mL^−1^.

### 4.2. Crystallization and Data Collection

Initial crystals were produced on the hanging-drop plate by equilibrating a mixture containing 1 μL of a protein solution (10–11 mg mL^−1^ protein in 20 mM Tris at pH 8.0, 150 mM NaCl, and 5 mM CaCl_2_) and 1 μL of a reservoir solution containing 10% polyethylene glycol 8000 (v/w), 100 mM magnesium acetate tetrahydrate, 200 mM potassium chloride, and 50 mM sodium cacodylate trihydrate, pH 6.5, against 0.4 mL of reservoir solution. The best diffractable crystal of the TG2/Ca^2+^ complex was obtained by optimization in a buffer containing 6% polyethylene glycol 8000 (v/w), 80 mM magnesium acetate tetrahydrate, 100 mM potassium chloride, 0.5 mM CaCl_2_, and 100 mM sodium cacodylate trihydrate pH 6.8.

For data collection, the crystals were transiently soaked in reservoir solution supplemented with 30% (v/v) glycerol as a cryoprotectant. The soaked crystals were then frozen in liquid nitrogen. A diffraction data set was collected using beamline SB-II (5C) at the Pohang Accelerator Laboratory (PAL), South Korea. The data sets were indexed and processed using HKL2000 [38].

### 4.3. Structure Determination and Analysis

The structure was determined by a molecular replacement (MR) phasing method using Phaser [39]. The previously solved GTP-bound TG2 structure (PDB code 4PYG) was used as the search model. The initial model was further improved using Coot [40] and phenix.refine in Phenix [41]. The quality of the model was validated using MolProbity [42]. All the structural figures in this paper were generated using PyMOL [43].

### 4.4. CD Analysis

The overall secondary structures were measured by CD spectroscopy using a J-715 spectropolarimeter at the Korea Basic Science Institute in South Korea. The spectra were obtained from 200 to 260 nm at 25 °C in a quartz cuvette with a 0.1 cm pathlength using a bandwidth of 1.0 nm, speed of 50 mm/min, and response time of 5 s. The protein samples in the buffer containing 20 mM Tris-HCl at pH 8.0, 150 mM NaCl, and 5 mM CaCl_2_ were diluted to 0.1 mg/mL prior to use. Three scans were accumulated and averaged.

### 4.5. Protein Data Bank Accession Code

Coordinates and structure factors have been deposited in the Protein Data Bank under the PDB ID code, 6KZB.

### 4.6. ITC

ITC experiments were performed using a NanoITC (TA Instruments). TG2 purified without any ions including Ca^2+^ was dialyzed against PBS buffer and was used as the fixed sample. Various concentration of CaCl_2_ and MgCl_2_ were dissolved in the same buffer to minimize the heat of dilution values and were used for titrating samples. Prior to titration, the protein sample and ion samples were centrifuged at 13,000 rpm at 4 °C for 5 min to remove any precipitants. For incremental injection in ITC, 2 μL of the ion sample was injected into a sample cell containing 190 μL of TG2 at a concentration of ~20 μM. All titrations were carried out at 15 °C with 25 injections at 160 s intervals. The area under each titration peak was integrated, plotted against the number of injections, and fitted to a one-site independent binding model, using the software provided by TA instruments. Experimental data were subtracted from the appropriate baselines acquired by injecting ions into the buffer in a sample cell without the protein sample.

### 4.7. In Vitro Transamidase Activity Assay

Transamidase activity was measured in the solution phase, based on measuring the incorporation of BP (ThermoFisher, Waltham, MA) into *N,N*′-dimethylcasein (Sigma, St. Louis, MO, USA), as described previously [44], with the following modifications. Transamidase activity was determined using 10 nM of purified human TG2 WT and mutants at various concentrations of CaCl_2_ for 1 h at 37 °C. Mg^2+^ inhibition assay was performed in the presence of 2 mM CaCl_2_. A GTP inhibition assay was performed in the presence of 2 mM CaCl_2_ and 1 mM MgCl_2_.

### 4.8. GTP Binding Assay

GTP binding studies were performed using the GTP analogue, BODIPY FL GTP-γ-*S* (Molecular Probes, Eugene, OR, USA). The TG2 WT and mutants (1 μM) were incubated in the reaction solution containing 1.6 μM BODIPY FL GTP-γ-*S*, 2 mM DTT, 1 mM EDTA, 50 mM Tris/HCl, pH7.5, and various concentrations of MgCl_2_ (0–4 mM) for 20 min at room temperature. GTP binding affinity was determined by measuring the fluorescence intensity on an Infinite M200 system (excitation at 485 nm and emission at 520 nm; Tecan, San Jose, CA, USA).

### 4.9. GTPase Activity Assay

GTPase activity of TG2 was measured using the ATPase/GTPase Activity Assay Kit (Sigma, St. Louis, MO) as described previously [44]. In brief, 0.6, 1.2, and 2.5 μM of purified human TG2 and its mutants were incubated with a reaction solution containing 1 mM GTP and 4 mM MgAc_2_ for 2 h at 37 °C. The reaction was visualized by adding malachite green reagent solution. GTPase activity was quantified by measuring absorbance at 620 nm using a microplate spectrophotometer (Molecular Devices, Sunnyvale, CA, USA). The standard curve was generated with a phosphate solution.

### 4.10. Cell Culture

HDF, obtained from human foreskins, were purchased from Gibco (ThermoFisher, Waltham, MA, USA). HDF, HCT116, and HEK293 cells were grown in DMEM (LM001-05, Welgene, Gyeongsan-si, Korea) supplemented with 10% fetal bovine serum (Hyclone, GE healthcare, Pittsburgh, PA, USA), 100 units/mL penicillin, and 100 μg/mL streptomycin (ThermoFisher, Waltham, MA, USA). To test the effect of Mg^2+^ depletion on intracellular TG2 transamidase activity, cells were cultured in Mg^2+^-free DMEM (LM001-54, Welgene, Korea). For overexpression of TG2, pcDNA3 plasmids containing WT, E437R, E539R, E437/539R, and R580A of TG2 encoding V224 were used. TG2 mutants were generated by using the QuikChange site directed mutagenesis kit (Stratagene, Sigma, St. Louis, MO, USA). HEK293 cells were transfected using Lipofectamine 3000 (Invitrogen, ThermoFisher, Waltham, MA, USA) according to manufacturer’s manual, and were incubated for 48 h.

### 4.11. Intracellular Transamidase Activity Assay

The intracellular transamidase activity assay was measured as described previously [45]. Briefly, cells were incubated with 1 mM BP for 1 h. Cells were harvested in PBS supplemented with a protease inhibitor cocktail (Roche), followed by sonication for lysis and centrifugation (13,000 × *g*, 15 min, 4 °C). For the microtiter plate assay, 10 μg of cell lysates were added to each well of a 96-well microtiter plate (Nunc, ThermoFisher, Waltham, MA, USA), and were incubated overnight at 4 °C. The transamidase activity was determined by the amount of BP incorporated in cellular proteins by detection with horseradish peroxidase-conjugated streptavidin (Zymed, ThermoFisher, Waltham, MA, USA) at room temperature for 1 h, and were quantified by color reaction with o-phenylenediamine dihydrochloride (Sigma, St. Louis, MO, USA). After incubation for 5 min, the reaction was stopped by adding H_2_SO_4_. Absorbance at 490 nm was measured on a microplate spectrophotometer.

### 4.12. Luciferase Reporter Assay

To measure NF-κB activity, we generated HEK293 cells stably expressing NF-κB responsive luciferase reporter (pGreenFire1™-NF-κB-Puro-vector, System Biosciences, Mountain View, CA, USA). The cells were transfected with pcDNA3 plasmids containing TG2 WT (V224) or mutants. Cells were lysed in luciferase assay buffer (Promega, Madison, WI, USA) and were analyzed for luciferase activity on an Infinite M200 (Tecan, San Jose, CA, USA).

### 4.13. Western Blot Analysis

Western blot analysis was carried out as described previously [28]. Antibodies against the following proteins were purchased from the indicated company: P-p65 and p65 (Cell signaling, Danvers, MA, USA); Actin (Santacruz, Dallas, TX, USA). Monoclonal antibody against TG2 was prepared as described previously [5].

### 4.14. Statistical Analysis

Data were analyzed using multiple *t*-tests for comparisons of various data sets in a graph. Data were also analyzed using one- or two-way ANOVA with a Tukey’s post hoc test for multiple comparisons. Error bars are expressed as the mean ± SEM or as the mean and 95% confidence intervals (CI). All statistical analyses were performed using GraphPad Prism software.

## Figures and Tables

**Figure 1 ijms-21-00791-f001:**
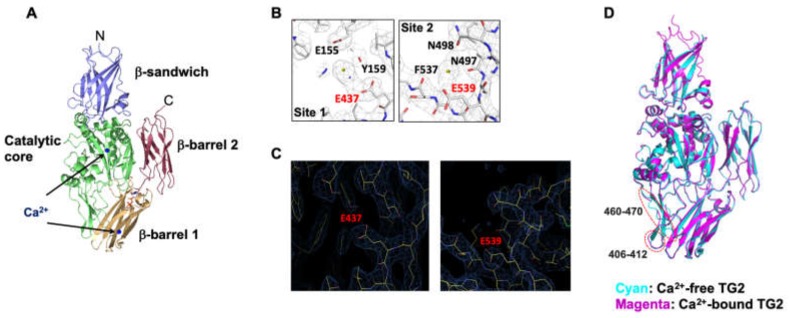
Three-dimensional structure of human transglutaminase 2 (TG2) with calcium ions (Ca^2+^). (**A**) Overall structure of the TG2/Ca^2+^ complex. Four distinct domains of TG2 are indicated with different colors. The locations of two Ca^2+^ are indicated in the cartoon figure. (**B**) An omit electron density map contoured at the 1-σ level around Ca^2+^. The Ca^2+^ coordinating residues are shown. (**C**) Absence of the electron density of Ca^2+^ in the Ca^2+^ binding sites of the apo-TG2 structure. (**D**) Superposition of the structure of the TG2/Ca^2+^ complex with apo-TG2 structure (PDB ID: 1KV3). Structurally different regions are indicated by red circles.

**Figure 2 ijms-21-00791-f002:**
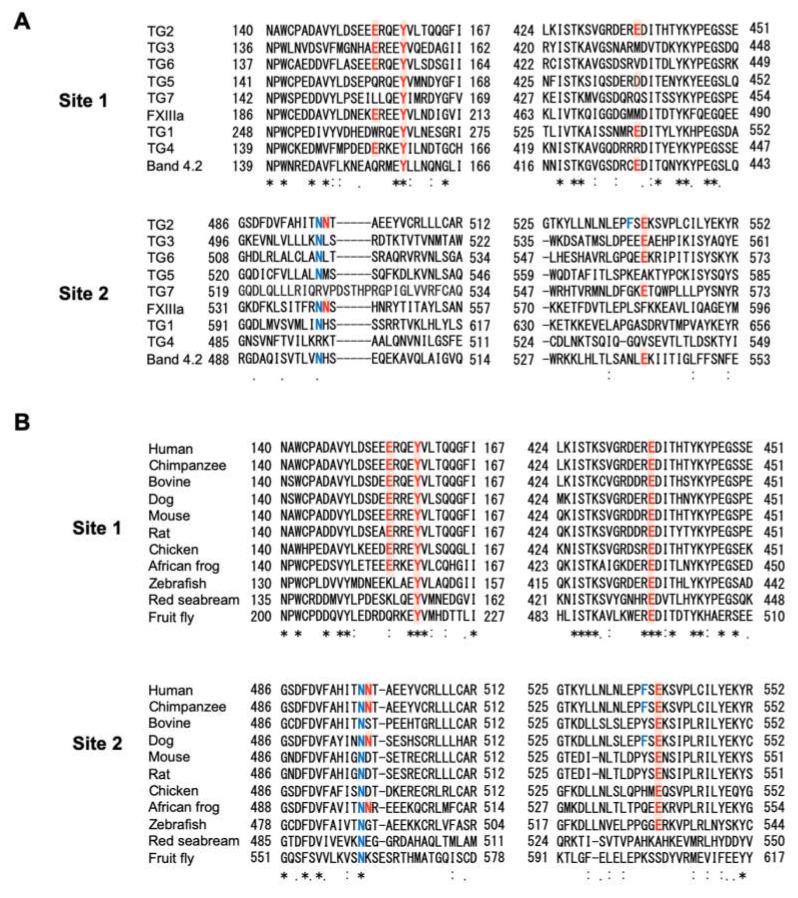
Multiple sequence alignment of the Ca^2+^-binding sites in transglutaminases. (**A**) Inter-isotype comparison of the human TG family. (**B**) Inter-species comparison of TG2. The amino acid sequence of TG2 was analyzed using Clustal Omega. Conserved Ca^2+^-binding amino acid residues (red characters) and similar amino acid residues are highlighted in red. Ca^2+^-binding amino acids in main chains are shown as blue characters. Asterisk (*), a conserved residue; Colon (:), a strongly similar residue; Period (.), a weakly similar residue.

**Figure 3 ijms-21-00791-f003:**
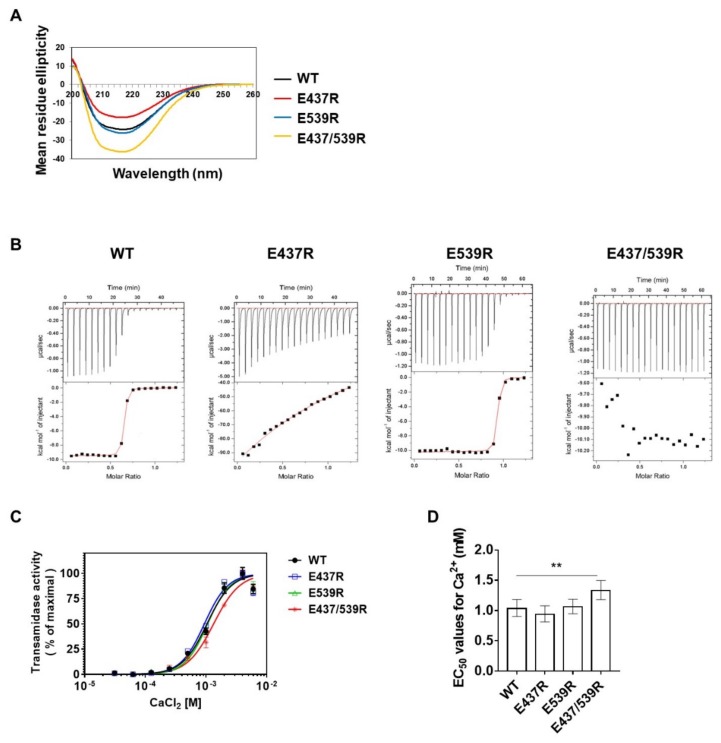
Binding of Ca^2+^ to E437 and E539 is required for the transamidase activity of TG2. (**A**) CD spectra of recombinant TG2 proteins. (**B**) Comparison of Ca^2+^-binding affinity between wild-type, E437R, E539R, and E437/539R using ITC. (**C**) Ca^2+^-dependent transamidase activities of TG2. The transamidase activities were measured in various concentrations of CaCl_2_ as described in Materials and Methods. (**D**) EC_50_ values for Ca^2+^ were calculated using GraphPad Prism software. Error bars are 95% CI of the EC_50_. One-way ANOVA with a Tukey post hoc test was used; ** *p* < 0.01.

**Figure 4 ijms-21-00791-f004:**
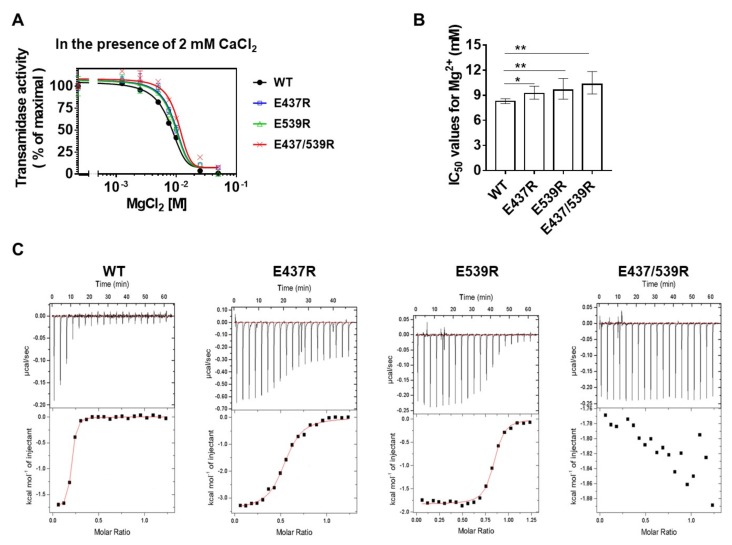
Binding of Mg^2+^ to E437 and E539 inhibits Ca^2+^-dependent transamidase activity of TG2. (**A**) Inhibition of transamidase activities of wild-type, E437R, E539R, and E437/539R by increasing Mg^2+^. Transamidase activities were measured in various concentrations of MgCl_2_ as described in Materials and Methods. (**B**) IC_50_ values for Mg^2+^ were calculated using GraphPad Prism software. Error bars are 95% CI of the IC_50_. One-way ANOVA with a Tukey post hoc test was used; * *p* < 0.05, ** *p* < 0.01. (**C**) Comparison of Mg^2+^-binding affinity of the wild-type and mutant TG2 using ITC.

**Figure 5 ijms-21-00791-f005:**
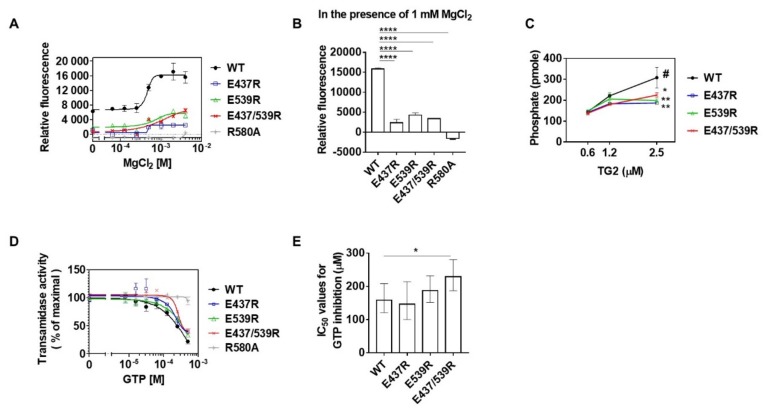
Binding of Mg^2+^ to E437 and E539 promotes GTP binding and hydrolysis activity of TG2. (**A**,**B**) Effect of Mg^2+^ on GTP-binding activities of the wild-type, E437R, E539R, and E437/539R. GTP binding activity was assessed using BODIPY FL GTP-γ-S, a GTP analogue, which fluoresces upon binding to proteins. The TG2 proteins were incubated with BODIPY FL GTP-γ-S at various concentrations of MgCl_2_. Fluorescence intensities were plotted against the MgCl_2_ concentration (**A**) and were selected at 1 mM MgCl_2_ for the bar graph (**B**). A GTP-binding defective R580A mutant was used as the negative control. One-way ANOVA with a Tukey post hoc test was used. Data represent the mean ± SEM; **** *p* < 0.0001. (**C**) GTP hydrolytic activity of wild-type and mutant TG2. Activity was determined by measuring the amount of phosphates released from GTP. Two-way ANOVA with a Tukey’s multiple comparisons test was used. Data represent the mean ± SEM; * *p* < 0.05 and ** *p* < 0.01 compared with the wild-type (#) at 2.5 μM TG2. (**D**) Inhibition of transamidase activities of TG2 by GTP. The transamidase activities were measured in various concentrations of GTP in the presence of 2 mM CaCl_2_ and 1 mM MgCl_2_ as described in Materials and Methods. (**E**) The IC_50_ values for GTP were calculated using GraphPad Prism software. One-way ANOVA with a Tukey post hoc test was used. Error bars are 95% CI of the IC_50_; * *p* < 0.05.

**Figure 6 ijms-21-00791-f006:**
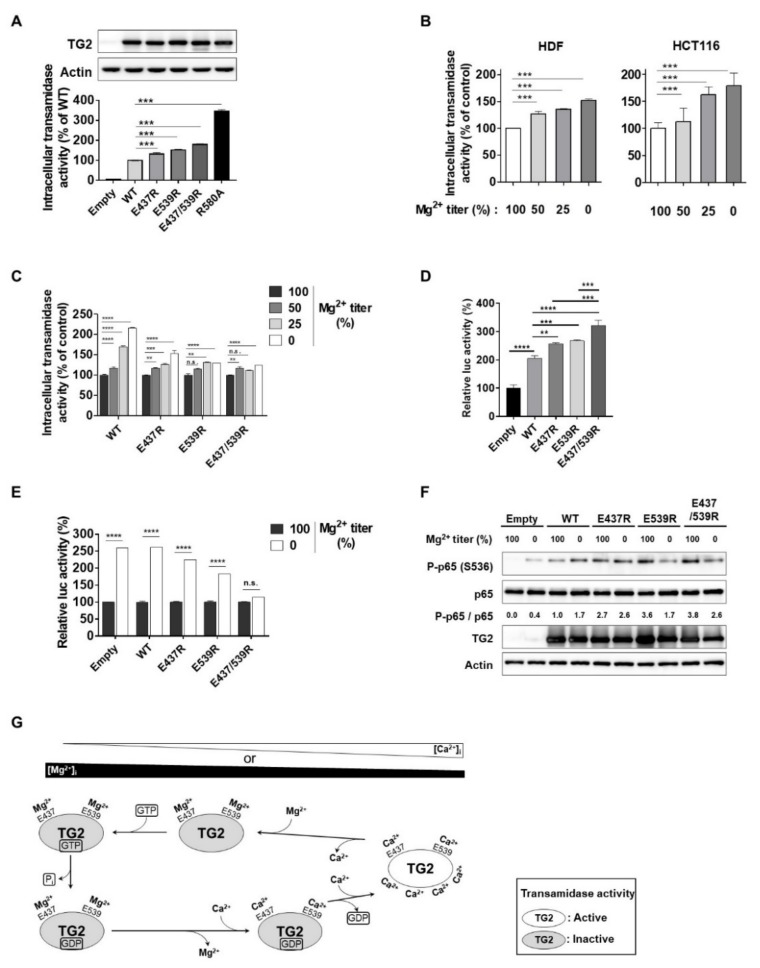
Binding of Mg^2+^ to E437 and E539 is critical for preventing the aberrant activation of intracellular TG2. (**A**) HEK293 cells overexpressing the wild-type, E437R, E539R, or E437/539R TG2s were cultured under normal conditions. (**B**) Effect of Mg^2+^-depletion on intracellular transamidase activities of HDF and HCT116 cells. (**C**) The wild-type, E437R, E539R, or E437/539R were overexpressed in HEK293 cells. Cells were cultured under normal or Mg^2+^-deprived conditions. Intracellular transamidase activities were analyzed as described in Materials and Methods. (**D**) The NF-κB luciferase reporter assay was performed in HEK293 cells overexpressing TG2. (**E**) HEK293 cells transfected with TG2 were incubated in normal or Mg^2+^-depleted culture media. NF-κB luciferase reporter activity was measured in the respective cell lysates. (**F**) Levels of phosphorylated p65 (Ser-536), p65, TG2, and actin were assessed by immunoblotting with the specific antibodies. One-way ANOVA with a Tukey post hoc test (**A**–**D**) or multiple *t*-tests (**E**) were used; ** *p* < 0.01, *** *p* < 0.001, **** *p* < 0.0001; n.s., not significant. (**G**) A model summarizing the allosteric regulation of TG2 by competitive binding of Ca^2+^ and Mg^2+^. Under normal conditions, intracellular TG2 is bound to Mg^2+^ at E437 and E539, which promotes GTP binding and hydrolysis, thereby inhibiting the transamidase activity. When (Mg^2+^)_i_ decreases or (Ca^2+^)_i_ increases, TG2 gets dissociated from Mg^2+^ and GTP or GDP, leading to increased transamidase activity.

**Table 1 ijms-21-00791-t001:** Crystallographic statistics.

Data Collection	Native
X-ray source	Synchrotron (PAL 5C)
Detector	Eiger 9M
Wavelength	1.0000
Space group	*C222_1_*
Cell dimensions	
*a*, *b*, *c*	133.1 Å, 216.3 Å, 166.3 Å
Resolution	50–3.56 Å
Wilson B-factor	80.666 Å^2^
†No. of unique reflections overall	29,265
† *R*_sym_	9.4% (34.5%)
†*I*/*I*	19.2 (3.9)
†Completeness	100% (99.9%)
†Redundancy	10.9 (11.0)
Refinement	
Resolution	42.58–3.55 Å
No. of reflections used (completeness)	27,758 (99.59%)
†*R*_work_	22.5% (22.68%)
†*R*_free_	26.3% (26.85%)
Average B-factors	
Protein	61.0 Å^2^
Other small molecules	82 Å^2^
Root mean square deviations	
Bond lengths	0.013 Å
Bond angles	1.672°
MolProbity analysis	
Ramachandran outliers	0.00%
Ramachandran favored	98.00%
Ramachandran allowed	2.00%
Rotamer outliers	6.00%
Clashscore	18.00

† Highest resolution shell is shown in parenthesis.

**Table 2 ijms-21-00791-t002:** ITC affinity measurements of TG2 for Ca^2+^ and Mg^2+^.

Divalent Ions	WT *K*_d_ (μM)	E437R *K*_d_ (μM)	E539R *K*_d_ (μM)	E437R/E539R *K*_d_ (μM)
Ca^2+^	0.027 ± 0.004	188 ± 24	0.047 ± 0.001	NB
Mg^2+^	0.215 ± 0.056	1.459 ± 0.523	0.282 ± 0.023	NB

NB, no binding.

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
