# Peer review of "Competitive Binding of Magnesium to Calcium Binding Sites Reciprocally Regulates Transamidase and GTP Hydrolysis Activity of Transglutaminase 2"

_ijms, 2020, doi:10.3390/ijms21030791_

Round 1
Reviewer 1 Report
The study is of potential interest. The authors provide important new data on the role of Mg2+ in the regulation of of transglutaminase 2 and on its interplay with Ca2+ and GTP/GDP. This reviewer has only minor concerns.
The first sentence on page 2, lines 44-45, just like the last sentence in the discussion (page 11, lines 298-299) seem to be too far-reaching, omission or re-composition should be considered.
Page 2, the sentences starting on line 79 are somewhat obscure; more detailed explanation and clarification would help the readers in appreciating the topic and the results.
The author reference the study (ref. 12) in which five Ca2+ binding sites were reported. In the present study only one of them are demonstrated, plus an additional one is described. What about the other Ca2+ binding sites? The reason for the discrepancy should be discussed.
The authors constructed mutants in which E437 and/or E539 were replaced by Arg. It is not clear what is the rationale behind the replacement of amino acids with negatively charged side-chains by an amino acid with a bulky positively charged side-chain. The possible structural consequences were only partially explored.
The results provide an explanation on how TG2 remains inactive in intracellular environments. However, one would like to see a speculation on how the inhibitory effects become relieved during cell activationpathological conditions.
Certain experiments were carried out at 2 mM CaCl2 concentration, which is highly above the intracellular Ca2+. What is the physiological relevance of these experiments?
Author Response
Jan 15, 2020
We appreciate the reviewers’ valuable comments. We have revised the manuscript. The major changes made during revision are highlighted and are as follows:
(1) We omitted the first sentence on page 2, lines 44-45 in the latest version of the manuscript for revision (Comment 1 of reviewer #1).
(2) We added more detailed explanation in the sentence on line 79 in the latest version of the manuscript for revision (Comment 2 of reviewer #1).
(3) We added a paragraph in discussion for explanation of other calcium binding sites (Comment 3 of reviewer #1).
Followings are responses to other points of reviewer’s comments:
The reason for replacement of E437 and/or E539 with Arg in mutants (Comment 4 of reviewer #1):
è Ca2+ is also coordinated by side chains of Y159 and E155 in the site 1 and main chains of E497 and F537 in the site 2. Thus, we intended to completely interfere Ca2+ binding to these sites through replacement of E437 and E539 with positive charged Arg. As reviewer #1 commented, E437 and/or E539 mutations with Arg did not affect to folding properties of TG2 (Fig. 3A). For these reasons, we used E437R and/or E539R mutants in the present study.
The mechanism of TG2 activation during cell activation/pathological conditions, relieving Mg2+ inhibitory effects (Comment 5 of reviewer #1):
è During cell activation and pathological conditions, intracellular Ca2+ concentration increases. Ca2+ competes with Mg2+ for E437 and E539 sites. Given that TG2 showed about 8-fold higher affinity for Ca2+ than Mg2+ (Table 2), TG2 binds with Ca2+ more efficiently than Mg2+. In addition, intracellular Mg2+ is decreased in aging. Moreover, chronic Mg2+ deficiency is implicated in various pathogeneses including colorectal cancer, pancreatic cancer, coronary atherocalcification, and vessel stiffness. TG2 is presumed to be activated by Mg2+ depletion in these conditions. We summarized these in Fig. 6G.
The physiological relevance of 2 mM CaCl2 in Figure 4A (Comment 6 of reviewer #1):
è We used purified human TG2 (G224) to compare with Ca2+-free GDP binding TG2 crystal structure (PDB ID:1KV3), which has G224. TG2 (G224) requires higher Ca2+ concentration than the natural form (V224) for transamidase activation (PLoS One. 2018; 13(10) and Biochem J. 2013; 455(3):261–72). In our experimental system, purified recombinant human TG2 (G224) is maximally activated in the presence of 2 mM CaCl2 (PLoS One. 2018; 13(10)). Thus, we performed in vitro experiment in the presence of 2 mM CaCl2. On the other hand, we performed the overexpression experiments in cells using TG2 (V224). Therefore, our results demonstrate that TG2 is actually regulated by binding of Mg2+ and Ca2+ on E437 and E539 sites under physiological conditions.
I hope our new results and responses could convince reviewer #1.
Sincerely yours,
In-Gyu Kim, M.D., Ph.D.
Department of Biochemistry and Molecular Biology
Seoul National University College of Medicine
Reviewer 2 Report
In the manuscript entitled "Competitive binding of magnesium to calcium binding sites reciprocally regulates transamidase and GTP hydrolysis activity of transglutaminase 2", by Eui Man Jeong et al., the Authors identified two glutamate residues, E437 and E539, as Ca2+-binding sites required for the transamidase activity of TG2. In addition, they found that magnesium (Mg2+) competitively binds to the E437 and E539 residues enhancing the GTP binding/hydrolysis activity. So, they demonstrated that E437 and E539 are Ca2+-binding sites contributing to the reciprocal regulation of transamidase and GTP binding/hydrolysis activities of TG2 through competitive Mg2+ binding. The manuscript is well written and clear. However, some points need to be revised by the Authors, in particular:
In Fig.6A, blot is not representative of the densitometric analysis and the bands of the housekeeping protein (actin) is not good. Minor english grammar errors.Author Response
Jan 15, 2020
We appreciate the reviewers’ valuable comments. We have revised the manuscript.
Followings are responses to other points of reviewer’s comments:
In Fig.6A, blot is not representative of the densitometric analysis and the bands of the housekeeping protein (actin) is not good. (Specific point 1 of reviewer #2):
è The graph demonstrates the transamidase activity of TG2 (Fig. 6A, lower), not the TG2 expression levels calculated from western bands in Figure 6A upper panel.
Minor english grammar errors. (Specific point 2 of reviewer #2):
è We have extensively edited the English in our manuscript through Editage [http://www.editage.com]. We attach the Editing Certificate (Editing Certificate_ijms-697217.pdf).
I hope our new results and responses could convince reviewer #2.
Sincerely yours,
In-Gyu Kim, M.D., Ph.D.
Department of Biochemistry and Molecular Biology
Seoul National University College of Medicine